# Chronic Influence of Inspiratory Muscle Training at Different Intensities on the Serum Metabolome

**DOI:** 10.3390/metabo10020078

**Published:** 2020-02-21

**Authors:** Camila A. Sakaguchi, David C. Nieman, Etore F. Signini, Raphael M. de Abreu, Claudio D. Silva, Patrícia Rehder-Santos, Maria G. A. Carosio, Roberta M. Maria, Carla C. Dato, Heloisa S. S. de Araújo, Tiago Venâncio, Antônio G. Ferreira, Aparecida M. Catai

**Affiliations:** 1Physical Therapy Department, Federal University of São Carlos, São Carlos, SP 13565-905, Brazil; sakaguchicamila@estudante.ufscar.br (C.A.S.); etore8@gmail.com (E.F.S.); raphaelmartins.abreu@gmail.com (R.M.d.A.); claudio_acrobatica@hotmail.com (C.D.S.); rehderpaty@hotmail.com (P.R.-S.); 2North Carolina Research Campus, Appalachian State University, Kannapolis, NC 28081, USA; niemandc@appstate.edu; 3Chemistry Department, Federal University of São Carlos, São Carlos, SP 13565-905, Brazil; gabi_carosio@hotmail.com (M.G.A.C.); robmmaria@gmail.com (R.M.M.); t_venancio@yahoo.com (T.V.); giba_04@yahoo.com.br (A.G.F.); 4Nutrition course, Central Paulista University Center, São Carlos, SP 13563-470, Brazil; carladatonutricionista@gmail.com; 5Physiological Sciences Department, Federal University of São Carlos, São Carlos, SP 13565-905, Brazil; hsaraujo@ufscar.br

**Keywords:** breathing exercises, muscle strength, muscle endurance, inspiratory muscle, metabolomics, metabolism

## Abstract

This study investigated the chronic effect of inspiratory muscle training (IMT) on the human serum metabolome in healthy male recreational cyclists. Using a randomized, parallel group design, twenty-eight participants were randomized to three IMT groups: low intensity (LI, *n* = 7); moderate intensity (MI, *n* = 10); and high intensity (HI, *n* = 11). The IMT was performed for 11 weeks. Another group of participants under the same conditions, who did not perform the IMT but participated in all procedures, was included as controls (CG, *n* = 6). Blood samples were collected one week before and after 11 weeks of IMT and analyzed for metabolite shifts using 1H NMR. Statistical analysis included a 4 (group) × 2 (time) repeated measures ANOVA using the general linear model (GLM), and multivariate principal component analysis (PCA). Untargeted metabolomics analysis of serum samples identified 22 metabolites, including amino acids, lipids, and tricarboxylic acid cycle intermediates. Metabolites shifts did not differ between groups, indicating that IMT at three intensity levels did not alter the serum metabolome relative to the control group. These results reveal novel insights into the metabolic effects of the IMT and are consistent with the results from other studies showing negligible chronic alterations in the serum metabolome in response to physical training.

## 1. Introduction

During the past decade, metabolomics-based studies have provided new insights regarding the influence of exercise on metabolic, physiological, and health-related responses and adaptations [1,2]. Metabolomics may be an important tool for monitoring athletes during different phases of training [3]. A recent systematic review showed that most metabolomics-based studies have focused on the acute metabolite response to prolonged and intensive exercise, and that more information was needed regarding the chronic effects of exercise training [1].

Inspiratory muscle training (IMT) is a potential complementary system to improve sports performance [4,5,6]. In this method, an additional load is applied to the diaphragm and accessory inspiratory muscles to enhance strength and endurance [7]. IMT may attenuate the metaboreflex [6], in which the accumulation of lactate and other metabolites in the respiratory muscles [8] triggers a sympathetic response causing vasoconstriction in the exercise limbs and early exercise termination [5,9,10]. IMT may reduce perceived breathlessness, lactate accumulation, and peripheral fatigue, and accordingly improve exercise performance [4,5,7].

The physiological benefits of IMT in sports performance have been reported in several studies [4,5,7,10,11], but the underlying metabolic mechanisms remain unclear. This study was designed to investigate the effects of IMT performed at three intensities (low, moderate, and high) on the human serum metabolome of healthy male recreational cyclists. Untargeted metabolomics was performed using 1H NMR spectroscopy with advanced bioinformatics. We hypothesized that 11 weeks of IMT at moderate and high intensities would result in serum metabolite shifts that would improve scientific understanding regarding the underlying metabolic mechanisms.

## 2. Results

The analysis included 34 apparently healthy recreational cyclists (ages 20 to 40 years) who successfully adhered to all aspects of the study design. Baseline characteristics of the participants are summarized in Table 1. Participants did not differ in age, height, body mass, body mass index (BMI), body fat percentage, lean mass percentage, inspiratory muscle strength, or oxygen consumption at peak moment (VO_2_ peak). The food records revealed no difference over time in energy, macronutrient, and micronutrient intake during the study.

VO_2_ peak improved to a similar extent between the IMT groups (LI, MI, HI) after 11 weeks of training. Delta data (post-pre training) showed no difference between training groups (2.43 ± 6.19; 3.18 ± 5.54; 0.19 ± 6.56 ml·min^−1^·kg^−1^) (*p* = 0.520), nor in comparison to the control group (0.950 ± 4.06, *p* = 0.665). IMT improved inspiratory muscle strength (MIP) delta (post-pre training) in the LI, MI and HI training groups (25.9 ± 13.3; 44.7 ± 16.0; 55.6 ± 23.8 cmH_2_O) and in the control group (24.2 ± 16.2), but with significant differences in the HI group when compared to the LI group (*p* = 0.012) and to the control group (*p* = 0.009) (Figure 1).

A total of 22 metabolites (amino acids, fatty acids, tricarboxylic acid cycle (TCA) intermediates) were identified using 1H NMR spectroscopy (see Table 2 for the chemical shifts and peak areas). Repeated-measures ANOVA indicated no significant group x time interaction effects for all 22 metabolites (Table 2). These findings were confirmed by principal component analysis (PCA). The multivariate statistics results are depicted in the score plots in Figure 2. No valid model was found, indicating no significant separation in overall metabolite changes between the four groups. 

## 3. Discussion

This randomized, 11-week training study investigated the effects of IMT at three different intensities on the human serum metabolome. Untargeted metabolomics with 1H NMR identified 22 metabolites, including amino acids, lipids, and TCA intermediates. Contrary to our hypothesis, IMT at moderate and high intensities did not alter serum metabolite concentrations relative to the control group. This finding was confirmed through the use of bioinformatics procedures including PCA, in which no valid model was established that separated groups over the 11-week training period.

The magnitude of change in serum metabolites is directly related to exercise intensity and duration [12,13,14]. In acute exercise studies, prolonged and intensive exercise induces large-fold changes in a large number of metabolites including lipids, amino acids, and TCA cycle components [1]. Few studies have investigated the effects of chronic exercise training on the metabolome [14,15,16]. These studies included moderate and/or high intensity cycling and running protocols, and shifts in metabolites were limited and of low magnitude. Training-induced changes in serum metabolites lack consensus, but have been reported to include decreases in hippuric acid, lactate, pyruvate, and hypoxanthine, and increases in creatinine, dimethylamine, 3-methylxanthine, TCA cycle intermediates, and phospholipids [1].

IMT during an 11-week period improved muscle inspiratory strength but was not linked to significant metabolite shifts, and this finding is consistent with the few studies available. This study focused on a limited number of metabolites from bioenergetics pathways (amino acids, lipids, TCA) [17]. Comprehensive metabolomics-based studies using LC-MS/MS are needed to better define potential chronic metabolite shifts related to a variety of exercise training modalities including IMT. IMT can improve exercise performance, perhaps through alterations in metabolites linked to the metaboreflex [4,5,18] and cardiovascular autonomic modulation [10]. This effect may be captured through a more sensitive metabolomics platform than the one used in this study. This study was also limited by low participant numbers, in part due to the challenging time requirements of the training protocol [19]. The control group was not randomly selected, but was matched to training group participants for age and functional capacity.

IMT is frequently employed in the rehabilitation of individuals with chronic obstructive pulmonary disease (COPD) and heart failure [20,21,22,23,24,25]. In these groups, IMT improves inspiratory muscle strength, exercise capacity, cardiac and autonomic function, and quality of life [26]. An improvement in function and inflammatory biomarkers, including the reduction of tumor necrosis factor receptor 2 (sTNFR2) and an increase in adiponectin levels, was reported in hemodialysis patients after 8 weeks of IMT [27]. Metabolomics-based studies of IMT in diseased populations may improve the likelihood of measuring metabolite shifts related both to the disease process and to improvements in inspiratory muscle strength and exercise capacity.

In summary, the data of this metabolomics-based study indicated that IMT at three intensity levels did not alter the serum metabolome relative to the control group. These results are consistent with other exercise training studies showing minor alterations in the serum metabolome compared to the large but transient perturbations linked to prolonged and intensive exercise. Our findings provide important information for designing other future metabolomics-based studies investigating chronic effects of exercise. Additional research is needed to elucidate the metabolic adaptations induced by IMT.

## 4. Materials and Methods

### 4.1. Participants

Participants were recruited via mass advertising at the Federal University of São Carlos (UFSCar), local broadcast media, social networks and cycling groups in the city of São Carlos and the surrounding region. Participants included 34 apparently healthy recreational male cyclists. Inclusion criteria included the following: 20 to 40 years of age, nonsmokers, body mass index (BMI) less than 30 kg/m^2^, and regularly involved in cycling training (minimum practice of 150 min/week for at least 6 uninterrupted months before the experimental protocol). Participants voluntarily signed an informed consent form, and all study procedures were approved by the Human Research Ethics Committee at UFSCar (#2.303.309) and registered at ClinicalTrials.gov (#NCT02984189).

### 4.2. Research Design

This study utilized a longitudinal, randomized (1:1 allocation, using opaque envelopes), controlled, parallel group design. The methodological design was based on the Consolidated Standards of Reporting Trials (CONSORT) [28]. The full description of the protocol has been previously reported [29] and is graphically presented in Figure 2. IMT participants were randomly assigned to three groups: low intensity or sham group (LI, *n* = 7), moderate intensity (MI, *n* = 10), and high intensity (HI, *n* = 11). LI performed the IMT using 6 cmH_2_O resistance, MI trained using an intensity representing 60% of the MIP, and HI used ≈ 85%–90% of the MIP. A group of participants who did not perform the IMT but participated in all assessments and procedures was included as controls (CG, *n* = 6). As summarized in Figure 3 and Figure 4, data were analyzed from subjects (*n* = 34) who completed all aspects of the study. Blood samples were collected pre- and post-protocol at the 1st and 12th week.

### 4.3. Experimental Section 

All procedures of experimental tests and IMT were conducted at the Cardiovascular Physical Therapy Laboratory—Nucleus of Research in Physical Exercise (NUPEF) at the Department of Physical Therapy (DFisio). Clinical ergometry evaluations were performed at the Physical Therapy in Cardiology area of the Health School Unit (USE). Metabolomics data processing and analysis were performed at the Laboratory of Nuclear Magnetic Resonance of the Department of Chemistry at UFSCar.

#### 4.3.1. Baseline Testing 

One month before baseline testing, participants were given an individualized food list with dietary guidelines provided by a dietitian. Participants followed the dietary plan to minimize potential effects of food intake variance on the serum metabolome. Participants reported in a fasted state for the first day of baseline testing to the UNILAB (UNIMED Clinical Analysis Laboratory of São Carlos), and blood samples were taken from an antecubital vein with subjects in the seated position.

Participants then reported to the Cardiovascular Physical Therapy Laboratory for pre-study fitness testing. VO_2_ peak was measured during a graded exercise test, with workload increments calculated for each participant using the formula described by Wasserman et al. [30]. Metabolic measurements were collected using the ULTIMA MedGraphics (St. Paul, MN, USA) metabolic cart, and exercise was performed on a cycle ergometer (CORIVAL V3, Lode BV, The Netherlands). Data were processed using the software Breeze Suite 7.1, MedGraphics (St. Paul, MN, USA). Body composition was measured with dual-energy X-ray absorptiometry (DXA) (Discovery DXA System, Hologic, Marlborough, MA, USA). Inspiratory muscle strength (IMS) was measured during maximal inspiratory effort maneuvers using a digital manovacuometer (MVD-300, 179 Globalmed, Porto Alegre, Brazil). Testing for respiratory muscle endurance was performed using an incremental protocol with a linear inspiratory loading device (PowerBreathe, Ironman K5, HaB Ltd, UK). These data were used to determine the HI training load [29].

#### 4.3.2. Inspiratory Muscle Training (IMT)

Details of the training protocol have been published previously [29]. Briefly, the 11-week IMT protocol consisted of three 1-hour sessions per week organized as follows: 5 min warm-up using 50% of the training load, and then 3 sets of 15 min, with 1-minute intervals between sets. IMT was conducted using a linear inspiratory loading device (PowerBreathe, Ironman K5, HaB Ltd, UK). Except for the LI group, loads were adjusted according to reassessment tests after 4 w and 8 w of training.

The 3-d food records were analyzed using a computerized dietary assessment program for energy and macronutrient content (Avanutri® revolution; Avanutri Equipamentos de Avaliação Ltda., Três Rios, RJ, Brazil). Variation in macronutrient balance was limited to 10% between records.

### 4.4. Sample Preparation for NMR Analysis

Blood samples were collected in serum separator tubes (S-Monovette 4.9 ml, Sarstedt, Germany), allowed to stand at room temperature. Serum samples were centrifuged at 3000 RPM for 10 min, and then stored at −80 °C until analysis [31]. The samples were thawed shortly before the analysis. Extraction of the metabolites was performed as follows: 600 µL of methanol-d4 was added to 200 µL of sample, followed by stirring for 5 min, an ultrasonic bath for 5 min, and centrifugation at 13,000 RPM for 10 min. After centrifugation, 600 µL of supernatant was transferred to a 5-mm NMR tube.

### 4.5. NMR Data Acquisition and Metabolite Identification

Metabolomics analyses were performed in accordance with published protocols [31]. Each spectrum was acquired using a Bruker Advance III 600 MHz spectrometer (Bruker Biospin, Germany) (operating at 600.08 MHz for 1 h) equipped with a 5-mm cryogenic probe and operating at a temperature of 23 °C. Spectra were manually phased and baseline-corrected using Bruker Topspin 3.5 software (Bruker Biospin, Germany). One-dimensional (1D) 1H NMR measurements were performed using the noesyprld pulse sequence (Bruker standard), with pre-suppression for water signal. A total of 64 scans (ns) were performed in 128-k data points over a spectral width of 15.0182 ppm, using a relaxation delay of 1 s and an acquisition time of 7.27 s. Two-dimensional nuclear magnetic resonance spectroscopy (2D NMR) was conducted and included correlation spectroscopy (COSY) and heteronuclear single-quantum correlation spectroscopy (HSQC) to confirm the assignment of peaks and identification of metabolites. After all spectra were acquired, phase adjustment, baseline correction, removal of water signal (4.6–5.1 ppm), spectral calibration and quantification were conducted following the parameters for profiling as defined in Chenomx NMR Suite software 8.31 (Chenomx Inc., Edmonton, Canada). Metabolites were identified using the internet database (Human Metabolome Database) and the software Chenomx NMR Suite, and results were compared with other publications [14,15].

### 4.6. Statistical Analysis 

Data were expressed as mean ± SD. Pre-study data and food record data were compared between groups (control, LI, MI, HI) using one-way analysis of variance (ANOVA). Statistical differences were accepted when the *p*-value was ≤ 0.05.

The serum metabolomics data were analyzed using the general linear model (GLM) and a 4 (group) × 2 (time) repeated-measures ANOVA (IBM SPSS Statistics for Windows, Version 24.0, IBM Corp, Armonk, NY, USA). Spectral data were analyzed by multivariate statistics using the software Amix® v 3.9.14 (Bruker BioSpin, Germany). PCA was performed using pre- and post-IMT data for all four groups. The PCA model considered 179 NMR spectra, 88 variable buckets and 4 principal components. The total explained variance was 54.70% for PC1vsPC2 and 95% confidence level. The scaling of rows in the bucket table was to total intensity and the columns to unit variance.

## Figures and Tables

**Figure 1 metabolites-10-00078-f001:**
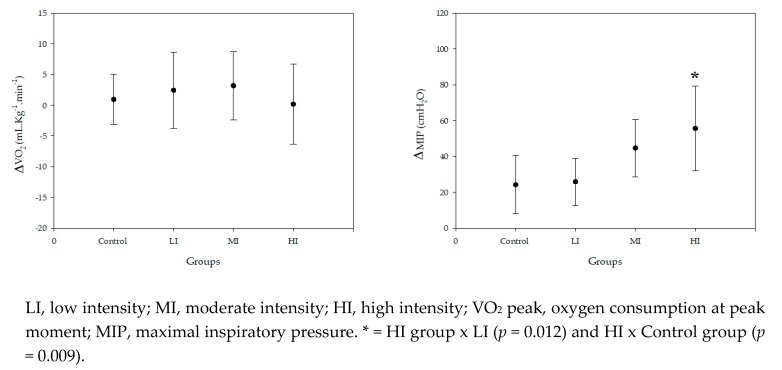
Delta data (post-pre protocol) of VO_2_ peak and MIP for the control, low intensity (LI), moderate intensity (MI), and high intensity (HI) groups.

**Figure 2 metabolites-10-00078-f002:**
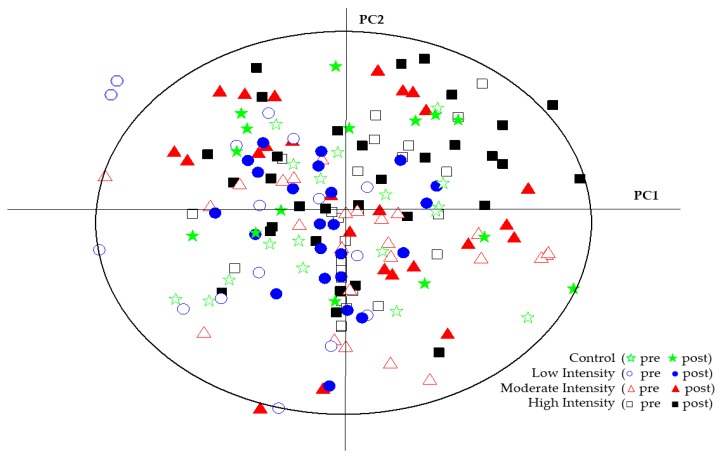
Principal component analysis (PCA) score plots of all serum samples pre- and post-protocol. Each color and shape represents a different training group and condition (described in the legend). No distinct separation between training groups pre- and post-protocol timepoints is observed.

**Figure 3 metabolites-10-00078-f003:**
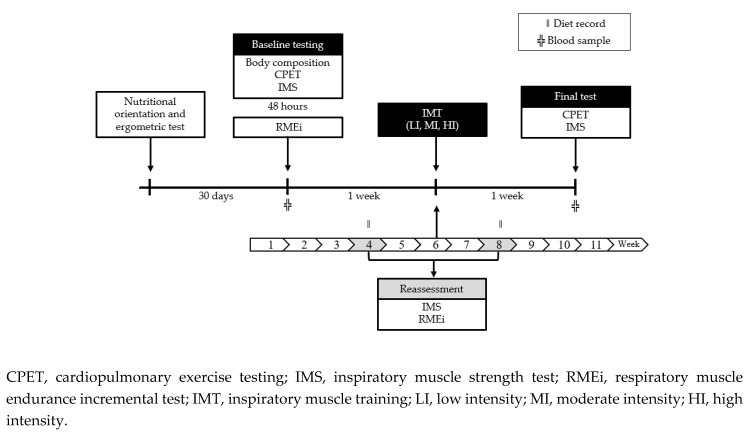
Protocol design. After a month following diet orientations, participants underwent 13 weeks of protocol. Training groups performed the inspiratory muscle training for 11 weeks. The control group participated in all assessments and procedures, including the reassessment tests and food records.

**Figure 4 metabolites-10-00078-f004:**
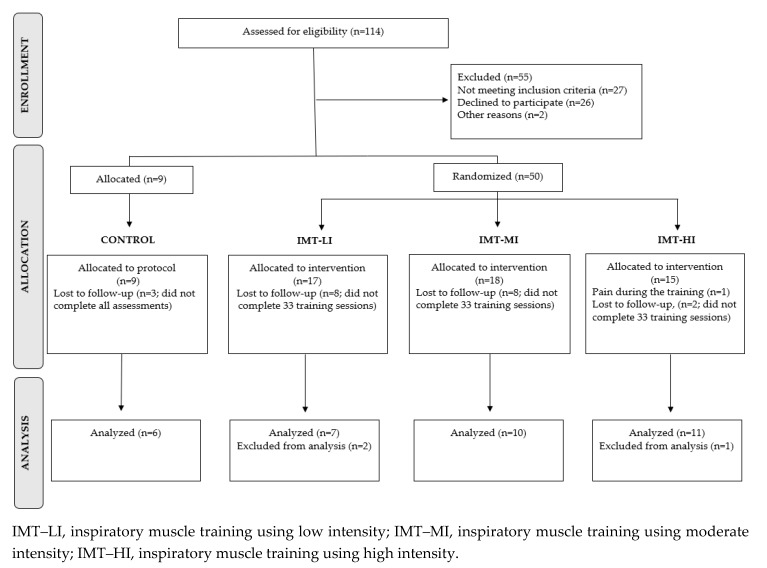
Flow diagram of participants.

**Table 1 metabolites-10-00078-t001:** Participants’ baseline characteristics (mean ± SD).

Variable	Control (*n* = 6)	LI (*n* = 7)	MI (*n* = 10)	HI (*n* = 11)	*P*-Value
Age (years)	30.8 ± 5.6	29.6 ± 4.9	31.9 ± 7.1	30.3 ± 7.2	0.896
Height (m)	1.74 ± 0.12	1.79 ± 0.03	1.75 ± 0.05	1.77 ± 0.06	0.580
Body mass (kg)	75.9 ± 18.0	75.9 ± 12.4	73.8 ± 8.2	78.5 ± 10.2	0.838
BMI (kg/m^2^)	24.9 ± 4.3	23.8 ± 4.4	23.9 ± 1.9	25.1 ± 3.4	0.804
Body fat (%)	22.0 ± 3.27	21.6 ± 6.31	21.1 ± 3.4	22.1 ± 4.72	0.958
Lean mass (%)	75.1 ± 3.52	74.9 ± 5.79	75.8 ± 3.25	74.8 ± 4.27	0.946
VO_2_ peak (ml.kg^−1^.min^−1^)	52.6 ± 15.2	40.9 ± 8.77	48.9 ± 9.60	48.8 ± 12.7	0.321
MIP (cmH_2_O)	151 ± 11.4	150 ± 10.4	159 ± 24.8	145 ± 13.7	0.303

LI, low intensity; MI, moderate intensity; HI, high intensity; BMI, body mass index; VO_2_ peak, oxygen consumption at peak moment; MIP, maximal inspiratory pressure.

**Table 2 metabolites-10-00078-t002:** Human serum metabolites identified by 1H NMR spectroscopy and peak areas in the control, low intensity, moderate intensity, and high intensity groups pre- and post-inspiratory muscle training.

Metabolites	Chemical Shift (ppm)	Control		Low Intensity		Moderate Intensity		High Intensity		*P*-Values: Interaction
		Pre-Training	Post-Training	Pre-Training	Post-Training	Pre-Training	Post-Training	Pre-Training	Post-Training	
Leucine	0.9	402 ± 41.8	425 ± 80.8	405 ± 52.1	400 ± 48.8	444 ± 73.0	388 ± 47.8	414 ± 46.1	428 ± 70.7	0.142
(CH_2_)n FA	1.26	1536 ± 180	1621 ± 316	1549 ± 192	1545 ± 195	1743 ± 290	1522 ± 202	1613 ± 168	1682 ± 296	0.144
Lactate	1.33	572 ± 104	598 ± 99.8	631 ± 110	642 ± 115	617 ± 120	574 ± 110	595 ± 111	595 ± 115	0.817
Alanine	1.47	47.4 ± 9.40	48.4 ± 4.88	53.8 ± 4.52	48.2 ± 16.7	48.0 ± 9.86	50.4 ± 10.05	52.1 ± 12.2	54.4 ± 10.7	0.540
CH_2_=CH_2_-CO lipids	1.6	251 ± 31.0	262 ± 46.8	249 ± 29.2	244 ± 34.5	267 ± 45.0	235 ± 31.5	254 ± 25.1	263 ± 41.7	0.216
Acetate	1.9	3.40 ± 1.09	4.44 ± 1.59	3.14 ± 1.04	3.08 ± 0.98	3.50 ± 1.20	3.13 ± 0.72	3.46 ± 1.66	2.79 ± 1.27	0.118
Acetone	2.2	8.97 ± 3.40	9.80 ± 2.97	9.14 ± 4.71	7.51 ± 1.81	28.1 ± 55.5	17.3 ± 31.5	16.6 ± 15.2	8.73 ± 5.07	0.924
Creatine	3.02	1.92 ± 0.90	1.97 ± 0.61	1.99 ± 0.69	1.91 ± 0.83	2.02 ± 0.50	2.18 ± 1.24	2.12 ± 0.89	2.68 ± 1.11	0.516
Creatinine	3.05	7.58 ± 2.06	7.27 ± 1.12	9.45 ± 1.93	8.05 ± 2.47	8.17 ± 1.31	7.76 ± 1.60	7.97 ± 0.99	8.22 ± 1.56	0.404
Arginine	3.19	64.4 ± 7.53	72.5 ± 11.8	66.4 ± 13.1	61.5 ± 11.6	64.6 ± 11.8	63.5 ± 14.5	68.0 ± 12.8	71.7 ± 13.3	0.264
Betaine	3.25	119 ± 20.8	137 ± 18.8	122 ± 24.3	120 ± 16.7	134 ± 23.3	121 ± 16.2	128 ± 22.9	129 ± 33.8	0.137
Glycine	3.7	63.2 ± 6.82	71.0 ± 11. 7	67.8 ± 10.8	64.4 ± 11.8	64.7 ± 10.3	64.4 ± 13.6	68.4 ± 10.1	70.1 ± 12.0	0.378
Proline	4.2	50.3 ± 8.10	54.5 ± 10.1	51.2 ± 5.79	51.0 ± 5.53	55.3 ± 9.81	51.1 ± 6.56	52.8 ± 5.96	55.2 ± 11.0	0.342
Glucose	4.5	82.5 ± 11.8	95.7 ± 16.4	88.3 ± 16.0	83.9 ± 15.8	83.7 ± 13.4	83.2 ± 17.8	91.0 ± 13.7	91.8 ± 16.0	0.270
Glucose	5.2	62.0 ± 8.02	70.1 ± 11.4	67.2 ± 10.6	63.7 ± 12.5	64.1 ± 10.3	63.6 ± 13.1	68.1 ± 10.2	70.0 ± 11.9	0.352
FA=CH	5.3	342 ± 36.3	366 ± 74.0	349 ± 52.6	346 ± 49.1	387 ± 63.9	349 ± 49.5	369 ± 36.5	377 ± 66.4	0.315
Urea	5.7	141 ± 31.1	162 ± 50.6	152 ± 34.4	136 ± 45.1	143 ± 40.4	126 ± 19.8	154 ± 36.7	145 ± 53.0	0.375
Tyrosine	6.8	4.46 ± 0.84	4.85 ± 0.62	4.67 ± 0.58	4.23 ± 1.09	4.46 ± 0.80	4.41 ± 0.88	4.56 ± 0.69	4.89 ± 0.95	0.417
Methyl histidine	7.0	2.42 ± 0.25	2.54 ± 0.39	2.60 ± 0.22	2.02 ± 0.95	2.52 ± 0.35	2.43 ± 0.40	2.54 ± 0.36	2.51 ± 0.35	0.155
Phenyalanine	7.4	2.09 ± 0.38	2.32 ± 0.24	2.24 ± 0.24	2.01 ± 0.62	2.21 ± 0.35	2.11 ± 0.35	2.06 ± 0.31	2.24 ± 0.36	0.105
Histidine	7.7	2.55 ± 0.19	2.73 ± 0.42	2.77 ± 0.20	2.18 ± 1.06	2.70 ± 0.39	2.63 ± 0.44	2.72 ± 0.39	2.69 ± 0.34	0.156
Tryptophan	7.72	2.10 ± 0.47	2.41 ± 0.28	2.37 ± 0.25	2.13 ± 0.50	2.13 ± 0.35	2.16 ± 0.31	2.13 ± 0.35	2.34 ± 0.34	0.056
Formate	8.5	0.31 ± 0.07	0.30 ± 0.04	0.27 ± 0.06	0.29 ± 0.04	0.30 ± 0.06	0.30 ± 0.08	0.30 ± 0.07	0.26 ± 0.06	0.512

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
