# Peer review of "Chronic Influence of Inspiratory Muscle Training at Different Intensities on the Serum Metabolome"

_metabolites, 2020, doi:10.3390/metabo10020078_

Round 1

Reviewer 1 Report

In the article with the title "Chronic influence of inspiratory muscle training at
3 different intensities on the serum metabolome" Sakaguchi et al. determine the serum metabolome after three different exercise training programs. Although the authors did not detect major changes as have been reported after acute exercise this manuscript still offers some intriguing insights to the response of metabolites to chronic exercise. For easier understanding, the authors should provide figures for their physiological readouts that are now just mentioned in the text (line 63 to 67). In addition for completeness the authors should also include the read outs of the control group .

Author Response

We appreciate the time and effort you made to review our paper. Please see responses to your comments below:

Point 1:  For easier understanding, the authors should provide figures for their physiological readouts that are now just mentioned in the text (line 63 to 67).

RESPONSE:

We added a figure (Figure 1. – line 70).  This figure illustrates the delta values post-pre training, and should improve the interpretation and comparison of physiological data between groups.

Point 2: In addition for completeness the authors should also include the read outs of the control group .

RESPONSE:

Data for the control group were added on lines 65, 67, 68 and 69.

Reviewer 2 Report

The authors conducted an untargeted metabolomics study to assess the effect of inspiratory muscle training (IMT) on the serum metabolome. Despite the overall negative observations, this research should propel future well-designed studies by improving the detection platform and sample size, and by including diseased populations to uncover the chronic effects of IMT, the metabolic adaptations to IMT, and the resulting physiological changes benefiting human physical fitness. The study is clearly written.

Specific comments:

Serum and plasma metablome has been used interchangeably in the text. Please standardise to serum metabolome. Could the authors organise a table summarising the comparison results pre-post training for VO2peak, MIP and other measurements? It will help the readers to better assess/interpret the results.  Could the authors modify the figure legends, by combining e.g. colour and shape to classify the groups in the PCA plot for better illustration? The authors have randomised the participants for the 3 training groups but not the control. Could the authors shed light on why the randomisation has not been done across the 4 groups and discuss the implications in "Discussion"?

Author Response

We appreciate the time and effort you made to review our paper. Please see responses to your comments below:

Point 1: Serum and plasma metablome has been used interchangeably in the text. Please standardise to serum metabolome.

RESPONSE:

Thank you, we corrected this and used "serum" throughout the paper.

Point 2: Could the authors organise a table summarising the comparison results pre-post training for VO2peak, MIP and other measurements? It will help the readers to better assess/interpret the results. 

RESPONSE:

We added a figure (Figure 1) to improve interpretation of the most important physiological outcomes related to IMT, MIP and VO2peak.   

Point 3: Could the authors modify the figure legends, by combining e.g. colour and shape to classify the groups in the PCA plot for better illustration?

RESPONSE:

The illustration was modified according to your recommendations.   
